# Photoswitchable polyynes for multiplexed stimulated Raman scattering microscopy with reversible light control

Yueli Yang[1], Xueyang Bai [1] & Fanghao Hu [1] ✉

Optical imaging with photo-controllable probes has greatly advanced biological research. With superb chemical specificity of vibrational spectroscopy, stimulated Raman scattering (SRS) microscopy is particularly promising for super-multiplexed optical imaging with rich chemical information. Functional SRS imaging in response to light has been recently demonstrated, but multiplexed SRS imaging with reversible photocontrol remains unaccomplished. Here, we create a multiplexing palette of photoswitchable polyynes with 16 Raman frequencies by coupling asymmetric diarylethene with super-multiplexed Carbow (Carbow-switch). Through optimization of both electronic and vibrational spectroscopy, Carbow-switch displays excellent photo-switching properties under visible light control and SRS response with large frequency change and signal enhancement. Reversible and spatial-selective multiplexed SRS imaging of different organelles are demonstrated in living cells. We further achieve photo-selective time-lapse imaging of organelle dynamics during oxidative stress and protein phase separation. The development of Carbow-switch for photoswitchable SRS microscopy will open up new avenues to study complex interactions and dynamics in living cells with high spatiotemporal precision and multiplexing capability.

Optical microscopy with high spatial and temporal photocontrol has played crucial roles in modern biological imaging, especially for fluorescence microscopy. The development of photo-controllable fluorescent probes with different colors has provided key tools to visualize selected molecules and processes at the cellular level with high specificity[1–3]. Moreover, photoactivatable and photoswitchable fluorescent probes have greatly facilitated the development of super-resolution nanoscopy, such as photoactivated localization microscopy (PALM)[4], stochastic optical reconstruction microscopy (STORM)[5], and reversible switchable/saturable optical fluorescent transitions (RESOLFT)[6,7], pushing the spatial resolution beyond optical diffraction limit.

Stimulated Raman scattering (SRS) microscopy, as an emerging nonlinear optical technique, has gained increasing attention in biological imaging[8–13]. With superb sensitivity and chemical specificity, SRS microscopy has demonstrated unique imaging capabilities complementary to fluorescence microscopy. Since Raman scattering is intrinsically rich in chemical information, label-free SRS imaging of many species including total protein and lipids has been achieved inside live cells, tissues and organisms[14–20]. With much smaller size than typical fluorescent probes, Raman tags of a few atoms such as alkyne have been developed to label specific small biomolecules such as metabolites and drugs, with minimal perturbation in cells[21–27]. Furthermore, the linewidth of Raman peak is nearly 50 times narrower than that of fluorescence transition, resulting in much higher multiplexing capability[28–32]. Two sets of Raman palettes based on xanthenes[28] and polyynes[29] (MARS and Carbow, respectively) with up to 20 frequency channels have been reported for super-multiplexed cellular imaging.

Multiplexed SRS imaging with selective photocontrol will enable functional imaging of various targets and regions with high spatial and

[1]Department of Chemistry, MOE Key Laboratory of Bioorganic Phosphorus Chemistry and Chemical Biology, Tsinghua University, 100084 Beijing, China.
✉e-mail: hufanghao@tsinghua.edu.cn

temporal precision. Photoactivatable Raman probes have been recently reported by uncaging cyclopropenone to generate the alkyne signal, and have been used for 3-color SRS imaging in live cells[33]. The photodecarbonylation process of cyclopropenone is irreversible, which limits photoactivation and selective imaging for one round. Dual-color photoswitchable SRS imaging has also been demonstrated using symmetric dithienylethene molecules[34,35]. Photo-induced cyclization of dithienylethene modifies the Raman frequency, which allows photoswitchable SRS detection. The molecular photoswitching properties, however, have not been systematically explored for SRS imaging, and multiplexed SRS imaging with reversible photocontrol remains unachieved.

Here, we develop multiplexed photoswitchable SRS imaging technique by coupling asymmetric diarylethene (DAE) structure with super-multiplexed Carbow palette. We design, synthesize a series of asymmetric DAE polyynes, and tune their electronic and vibrational spectra through heterocycle modification, polyyne conjugation, and isotope editing as well as end-capping substitutions. This generates photoswitchable polyynes with excellent photoswitching properties under visible light control and SRS signal with large frequency shifts and strong intensity enhancements. Moreover, we successfully obtain a new multiplexing vibrational palette with 16 photoswitchable Raman frequencies, which is termed as Carbow-switch. Through functionalization of Carbow-switch, we demonstrate multiplexed photoswitchable SRS imaging of specific organelles inside living cells. We further achieve region-selective and time-lapse imaging of subcellular dynamics during oxidative stress and protein phase separation, which reveals complex organelle organizations under cellular stress. Thus, Carbow-switch enables multiplexed SRS imaging of live cells with reversible photocontrol, which will be a powerful tool for studying subcellular interactions and dynamics with high spatiotemporal selectivity.

## Results

### Asymmetric diarylethene polyynes for photoswitchable SRS microscopy

To achieve efficient photoswitchable molecular detection, it is desired to have large frequency or intensity changes under light irradiations. Diarylethene is known to be a photoswitchable motif that undergoes reversible ring-cyclization reaction under UV or visible lights (Fig. 1a), which has shown excellent thermal stability and photo-fatigue resistance[36]. In addition, conversion from the open form to the closed form of DAE is accompanied by a red-shift in the absorption spectra from the UV to visible region. This is highly beneficial for electronic pre-resonance SRS (epr-SRS) excitation with signal amplification, when the absorption maximum is close to the near-infrared (near-IR) SRS wavelength[37]. We set out to synthesize a series of asymmetric DAE polyynes and optimized their electronic spectra in the near-IR range of 600-700 nm for sensitive SRS detection with large photoswitchable response (Fig. 1b).

With the design of asymmetric diarylethene, we can separately optimize both the electronic and vibrational spectroscopy of photoswitchable polyynes. Heterocycle incorporation is an effective strategy to tune the absorption spectra of diarylethenes. Asymmetric DAE was constructed by coupling thiophene, perfluorocyclopentene, and different heterocycles. Indeed, changing heterocycles from benzothiophene (1c-1) to 1-methylindole (2c-1) caused a large redshift of 72 nm in the closed forms from 562 nm to 634 nm (Fig. 1c and Supplementary Table 1). Substituting the indole ring at the 5' position with methoxy (3c-1) or dimethylamino (4c-1) groups can further redshift the absorption wavelengths to 675 nm and 724 nm, respectively (Fig. 1c). The large redshifts in the absorption maxima are due to the increased π conjugation and electron delocalization of indole ring and electron-donating groups, which lower the HOMO-LUMO gaps. Here, we

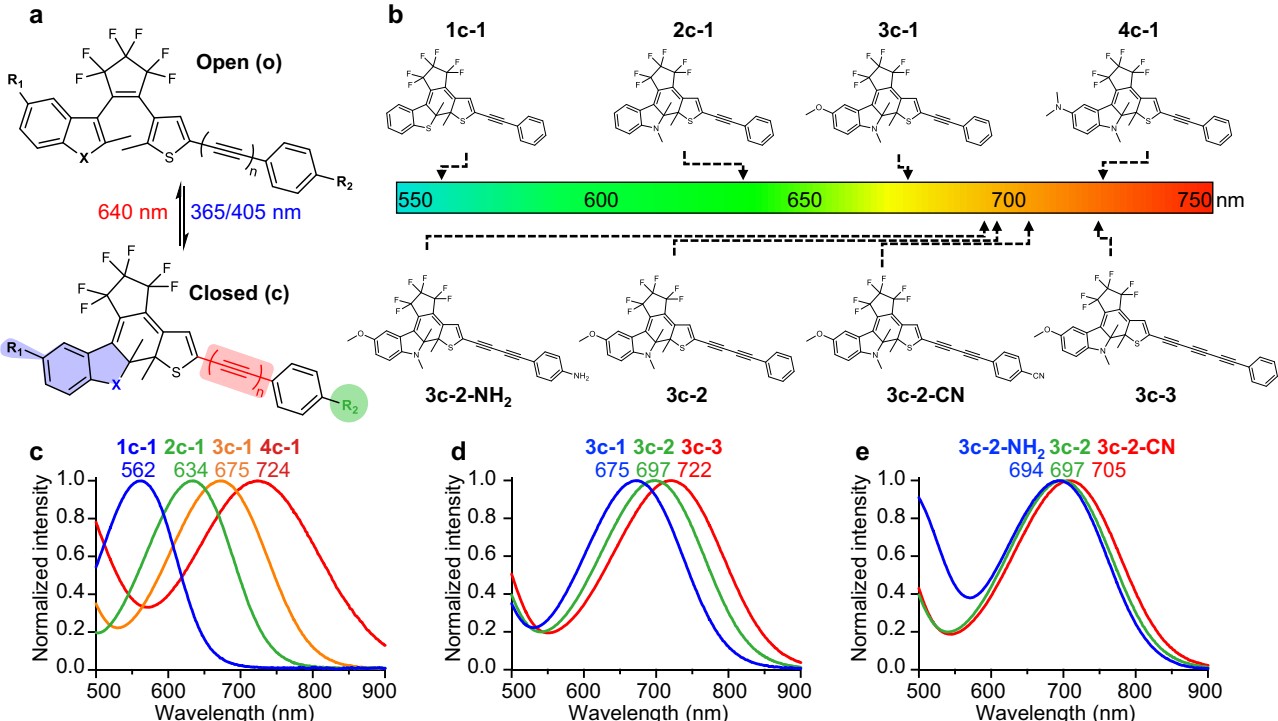

**Fig. 1 | Structural design and electronic spectra of asymmetric DAE polyynes. a** General structure of asymmetric DAE polyyne and reversible photo-conversion between the open and closed forms under light irradiations. Heterocycle (blue shaded area), polyyne (red shaded area), and end-capping substitution (green circle) were employed to tune the electronic spectra in the near-IR region. **b** Absorption wavelength optimization of asymmetric DAE polyynes in the closed forms. The absorption maxima indicated by arrows can be tuned from 560 nm to 720 nm. **c** Heterocycles variation from benzothiophene to 5-dimethylamino-1-methylindole caused large absorption redshifts of 160 nm. **d** Polyyne elongation from monoyne to triyne tuned the absorption maxima in the range of 50 nm. **e** End-capping substitutions from electron-donating -NH₂ to electron-withdrawing -CN group fine-tuned the absorption wavelengths within 10 nm.

denoted four heterocycles benzothiophene, 1-methylindole, 5-methoxy-1-methylindole, and 5-dimethylamino-1-methylindole by the first number **1**, **2**, **3**, and **4**, respectively. The second number represented the number of triple bonds in the polyyne conjugated to the thiophene, ranging from 1 to 4. And the letter in between represented the open (o) or closed (c) form. Thus, the absorption maxima of the closed forms can be redshifted on a large range of 160 nm from 560 nm to 720 nm through asymmetric heterocycle incorporation, pushing it into the near-IR region for epr-SRS excitation.

Moreover, polyynes were introduced to the asymmetric DAE scaffold independently to fine tune the absorption wavelengths and endow characteristic vibrational signatures in the cell Raman-silent region (1800-2700 cm[−1])[29,38]. All asymmetric DAE polyynes demonstrated good chemical stability under ambient condition when protected from light, as characterized by NMR. To our delight, incorporating polyynes into asymmetric DAE maintains the photo-cyclization properties (Supplementary Table 1). We showed that adding each triple bond resulted in a red-shift of ~25 nm in the absorption maxima, from 675 nm of monoyne 3c-1, to 697 nm of diyne 3c-2, to 722 nm of triyne 3c-3 (Fig. 1d), which is due to the successive elongation of π-conjugation system. Furthermore, end-capping substitutions of polyynes were used to tune the absorption wavelength within 10 nm. Electron-donating -NH$_2$ group (3c-2-NH$_2$) caused a blue-shift of 3 nm, while electron-withdrawing -CN group (3c-2-CN) can generate a bathochromic shift of 8 nm, compared to 697 nm of unsubstituted 3c-2 (Fig. 1e). Therefore, by combining heterocycle incorporation for coarse tuning, polyyne elongation and end-capping substitution for fine tuning, we have successfully tuned the near-IR electronic spectra of asymmetric DAE polyynes with high precision for photoswitchable SRS microscopy.

## Photoswitching behaviors of asymmetric DAE polyynes

With tunable absorption spectra, we characterized the photoswitching properties of asymmetric DAE polyynes. All asymmetric DAE polyyne compounds can undergo photo-cyclization under UV irradiation

(Supplementary Table 1), but 1-1 suffered from low conversion yield of 38% and 4c-1 showed a much slower conversion rate. While 2-1 and 3-1 have comparable photo-conversion yields, 3-1 displayed reduced extinction coefficient and broader absorption linewidth in the closed form, which could suffer from weaker epr-SRS enhancement and larger two-photon background[28]. Thus, with overall desirable photo-conversion yield and absorption profile, we chose 1-methylindole (**2**) as the optimal asymmetric DAE scaffold for further photoswitching studies (Supplementary Tables 1 and 2).

Take 2-3-OH as an example (Fig. 2a), photo-cyclization from 2o-3-OH to 2c-3-OH resulted in a large redshift in absorption peaks from 389 nm to 675 nm, with conversion yield as high as 91% measured by [1]H NMR (Fig. 2b, c). Between the open and closed forms, chemical shifts of three methyl protons of 2-3-OH were all upfield shifted (H1: Δδ = 0.92 ppm, H2: Δδ = 0.33 ppm, H3: Δδ = 0.15 ppm), allowing accurate quantification of conversion yield (Fig. 2c and Supplementary Fig. 1). Among all asymmetric DAE, we found that indole significantly enhanced the photo-conversion yield compared to benzothiophene, which can be further increased through polyynes elongation (Supplementary Table 1). Most asymmetric DAE polyynes showed high conversion yield (>70%), suggesting that indole and polyyne incorporations could be a general strategy to improve the photoswitching properties of DAE.

Photo-fatigue resistance of asymmetric DAE polyynes were also characterized. Due to the absorption redshifts in both the open and closed forms, not only it allowed epr-SRS excitation, but also 405 nm visible light can be used for photo-cyclization instead of 360 nm in previous study[34], which reduced UV photodamage. By alternating 405 nm and 640 nm illumination, we measured the absorption of 2-3-OH at 675 nm, which kept steady with little decay after 100 irradiation cycles (Fig. 2d and Supplementary Fig. 2). SRS intensity at 2147 cm[−1] was also used as the readout with consistent results (Supplementary Fig. 3), demonstrating excellent photo-fatigue resistance. Thus, asymmetric DAE polyynes can achieve reversible photoswitching with many cycles and high conversion yields.

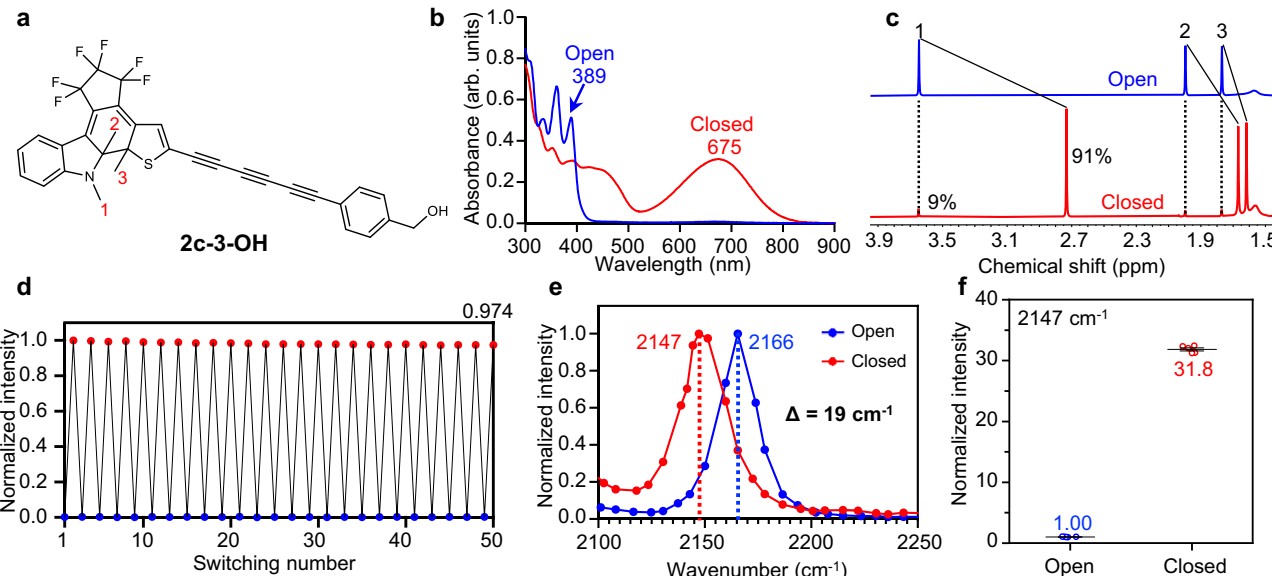

**Fig. 2 | Photoswitching properties and SRS spectroscopy of asymmetric DAE polyyne. a** Structure of 2-3-OH in the closed form. **b** UV-Vis absorption spectra of 2-3-OH in the open (blue) and closed (red) forms. Absorption peaks redshifted from 389 nm to 675 nm upon photo-cyclization. **c** [1]H-NMR spectra of 2-3-OH in the open (blue) and closed (red) forms, showing the upfield shifts in the methyl protons after photo-cyclization. Peaks 1, 2, and 3 of three methyl groups were indicated in **a**. Photo-conversion yield was quantified by methyl peak integration of two forms. **d** Reversible photoswitching of 2-3-OH over 25 switching cycles under alternating 405 nm and 640 nm illumination. Absorption at 675 nm was measured for photo-fatigue resistance characterization. **e** Normalized SRS spectra of 2-3-OH in both the open (blue) and closed (red) forms with a Raman frequency shift of 19 cm[−1]. **f** SRS signal ratio of 2c-3-OH over 2o-3-OH measured at 2147 cm[−1], with an intensity enhancement of 31.8 ± 0.5 (mean±s.d., $n$ = 5 measurements) upon photoswitching.

### SRS spectroscopy of asymmetric DAE polyynes

We further examined the photoswitchable SRS spectroscopy of asymmetric DAE polyynes. Between the open and closed forms, increased delocalized π-electrons enhance the electron-phonon coupling and soften the polyyne vibrational frequency[39]. Indeed, the Raman frequency of 2-3-OH decreased from 2166 cm⁻¹ (open-form) to 2147 cm⁻¹ (closed-form) with a large redshift of 19 cm⁻¹ after 405 nm irradiation (Fig. 2e), indicating strong vibrational coupling between polyyne and DAE. Additionally, the open-to-closed conversion of 2-3-OH resulted in a 32-fold increase in SRS signal when measured at the closed frequency of 2147 cm⁻¹ (Fig. 2f), which can be attributed to the effects of both the large frequency shift and the epr-SRS excitation. With strong signal enhancement, the highest SRS intensity of asymmetric DAE polyynes 2c-4-OH was nearly 500 times of 5-ethynyl-2′-deoxyuridine (EdU, Supplementary Table 2 and Supplementary Fig. 4), and SRS spectrum of 2o-4-OH can be measured at 10 μM with a good signal to noise ratio of 8 under 1 ms time constant (Supplementary Fig. 5). Based on the large photoswitchable signal response, we denote the open-to-closed conversion as on-switching and closed-to-open conversion as off-switching.

Using SRS microscope with high-speed detection, we quantified the photoswitching kinetics of 2-3-OH. The on-switching intensity reached plateau within 200 μs under low power of 405 nm irradiation (Supplementary Fig. 6). The photoswitching power dependence was also optimized for SRS imaging, and microwatt-level power (<1 mW) of either 405 nm or 640 nm illumination was sufficient to achieve high conversions of 2-3-OH (Supplementary Fig. 7). Therefore, asymmetric DAE polyynes display outstanding photoswitching behaviors including strong signal response with well-resolved frequencies and fast switching kinetics under low power of visible light, which are well suited for photoswitchable SRS imaging with high sensitivity and specificity.

### Multiplexing photoswitchable polyynes for SRS detection

Simultaneous visualization of different targets requires multiplexed imaging of distinguishable signal such as frequencies. Learned from the Carbow palette, we applied three approaches to expand the vibrational frequencies of photoswitchable polyynes for multiplexed SRS detection (Fig. 3a, b). First, end-capping group of polyyne was shown to modify the vibrational frequency significantly. On one hand, replacing the phenyl ring with alkyl group from 2-2-OH to 2-2-(CH₂)₂OH caused blueshifts of Raman frequencies from 2205 cm⁻¹ to 2232 cm⁻¹ in the open form and from 2190 cm⁻¹ to 2219 cm⁻¹ in the closed form (Fig. 3c, d and Supplementary Table 2). On the other hand, adding amino group at the para-position of the phenyl ring (2-2-NH₂) redshifted the Raman frequencies to 2194 cm⁻¹ (open) and 2170 cm⁻¹ (closed) due to the strong electron-donating effect, compared to unsubstituted 2-2-OH (Supplementary Table 2). Furthermore, changing the end-capping group to trimethylsilyl (TMS) group or hydrogen atom strongly decreased the Raman frequencies by 60 cm⁻¹ and 110 cm⁻¹ in both the open and closed forms (Supplementary Table 2), achieving wide vibrational frequency expansion.

Second, we applied ¹³C stable isotope labeling to polyyne triple bonds for additional frequency tuning. Vibrational frequency is inversely proportional to the square root of the reduced mass in a simplified harmonic oscillator model. Indeed, in both the open and closed forms, the Raman frequencies of 2-1-¹³C-H and 2-1-¹³C-TMS

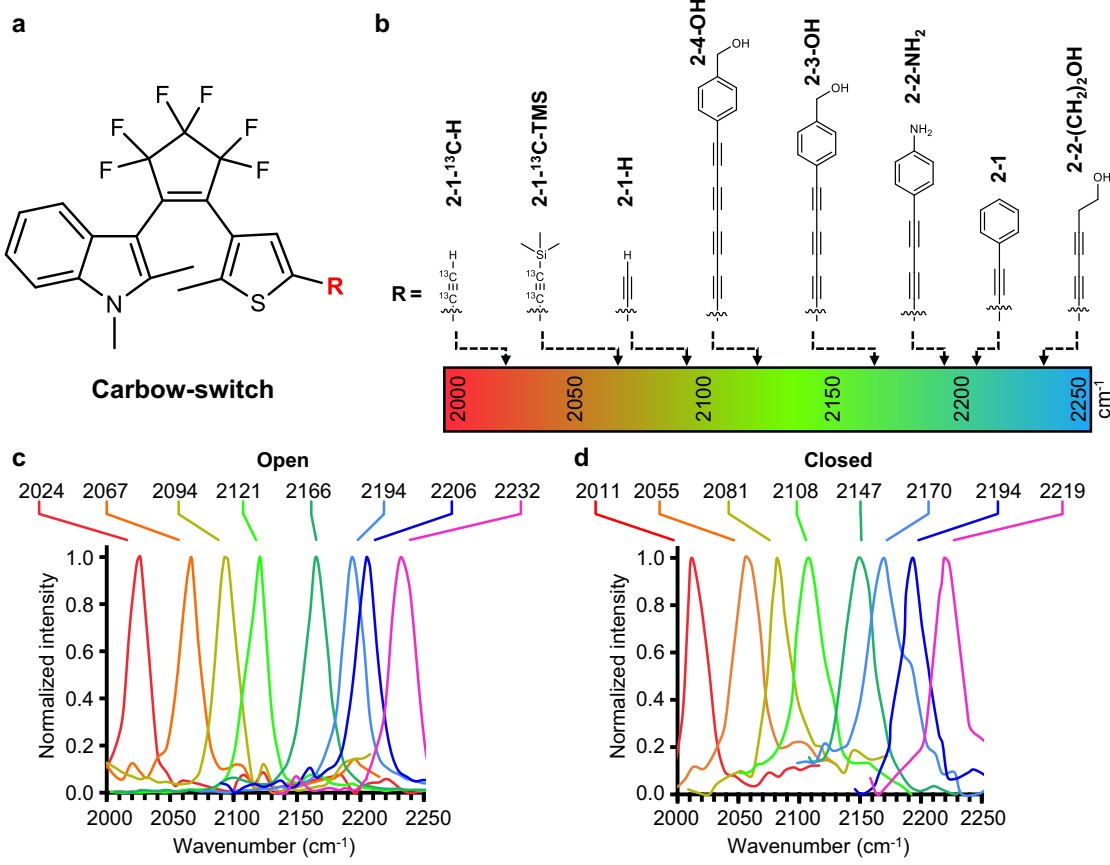

**Fig. 3 | Multiplexed photoswitchable polyynes. a** General structure of photoswitchable polyyne for vibrational multiplexing. **b** Chemical structures of eight Carbow-switch with distinct Raman frequencies through end-capping variation, ¹³C isotope labeling and polyyne elongation. **c, d** Sixteen normalized SRS spectra of Carbow-switch in both the open (**c**) and closed forms (**d**).

showed large redshifts of 70 and 80 cm$^{-1}$, respectively, compared to unlabeled 2-1-H and 2-1-TMS (Supplementary Table 2).

Third, polyyne elongation was applied not only to tune the near-IR absorption wavelength and enhance the photo-conversion yield, but also to soften the Raman frequency of the collective vibrational mode. From 2-2-OH to 2-3-OH to 2-4-OH, the Raman frequencies of the open forms decreased from 2205 cm$^{-1}$ to 2166 cm$^{-1}$ to 2121 cm$^{-1}$, respectively, and the closed form frequencies redshifted correspondingly from 2190 cm$^{-1}$ to 2147 cm$^{-1}$ to 2108 cm$^{-1}$ (Supplementary Table 2). In both the open and closed forms, each consecutive triple bond elongation can cause a frequency shift of ~40 cm$^{-1}$, due to the increased electron-phonon coupling in the polyynes[39,40]. The large frequency tuning in both the open and closed forms has greatly expanded the multiplexing channels, while maintaining well-resolved photoswitchable detection (Fig. 3c, d).

By combining end-capping variation, $^{13}$C isotope labeling, and polyyne elongation, we have created a palette of photoswitchable polyynes with 16 Raman frequencies for multiplexed SRS detection, which was termed as Carbow-switch (Fig. 3). Carbow-switch showed clearly distinguishable Raman frequencies in both open and closed forms with closest frequency separation of >12 cm$^{-1}$ and low spectral crosstalk (Fig. 3c, d). We noted that certain background signal was observed in the closed form, which was due to the two-photon absorption in the epr-SRS excitation. More importantly, all Carbow-switch molecules can be reversibly switched on/off under 405 nm or 640 nm visible light with frequency shifts ranging from 12 to 19 cm$^{-1}$, up to 24 cm$^{-1}$ in 2-2-NH$_2$ (Supplementary Fig. 8 and Supplementary Table 2). Due to both the frequency shift and epr-SRS excitation, Carbow-switch displayed large signal changes upon photoswitching, with intensity ratio as high as 90 folds (Supplementary Table 2). Therefore, Carbow-switch palette allowed photoswitchable SRS imaging with high signal response and multiplexing capability.

## Live-cell multiplexed SRS imaging with Carbow-switch

Organelles play crucial roles in regulating cellular activities. We functionalized Carbow-switch molecules with organelle-targeting groups for subcellular SRS imaging (Fig. 4 and Supplementary Table 3). Triphenylphosphonium cation is an effective mitochondria-targeting group with high affinity to the negative membrane potential of mitochondrial matrix[41]. Triphenylphosphonium was conjugated through a carbamate linker and a short ethylene glycol chain to 2-2-OH, which was termed as Mito-switch (Fig. 4a). Lyso-switch was obtained by introducing a dimethylamine group to 2-2-(CH$_2$)$_2$OH for lysosome targeting, which can be protonated and enriched inside acidic lysosomes[29] (Fig. 4b). PM-switch was developed using 2-3-OH by attaching a zwitterionic group containing an ammonium cation and a sulfonate anion for plasma membrane targeting through electrostatic interaction[42] (Fig. 4c). All three organelle-targeted Carbow-switch showed characteristic intensity patterns in mitochondria, lysosomes, and plasma membrane of living cells, with high co-localization to commercial fluorescent probes (Fig. 4a–c).

To achieve multiplexed SRS detection, we then imaged each organelle-targeted probe at three frequency channels with negligible spectral crosstalk (Supplementary Fig. 9). Additionally, organelle-targeted Carbow-switch probes showed little cytotoxicity in cell viability assays (Supplementary Fig. 10) and high photostability in both forms after continuous SRS imaging (Supplementary Fig. 11). With well-resolved frequencies and excellent biocompatibility, we combined three organelle-targeting Carbow-switch probes together with EdU to achieve 6-color and 9-color SRS imaging of live cells in both open and closed photoswitching forms, including nucleus, plasma membrane, mitochondria, lysosomes, total protein, and lipids (Fig. 4d and Supplementary Fig. 12, respectively). Thus, organelle-targeted Carbow-switch was successfully developed for multiplexed SRS imaging of live cells with high sensitivity, specificity, and biocompatibility.

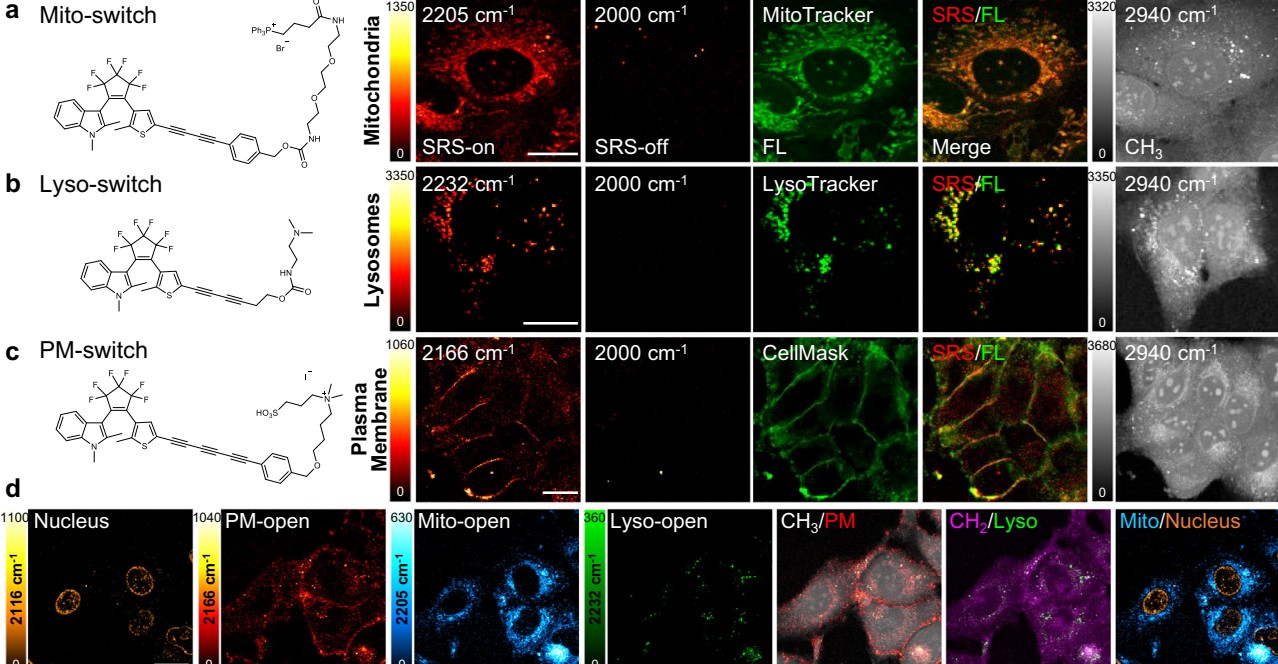

**Fig. 4 | Live-cell multiplexed SRS imaging of organelles with Carbow-switch.** **a**–**c** (Left) Chemical structures of organelle-targeted Carbow-switch. (Right) SRS images of corresponding Carbow-switch at the open forms in live U2OS cells, including Mito-switch (**a**, 2205 cm$^{-1}$), Lyso-switch (**b**, 2232 cm$^{-1}$), and PM-switch (**c**, 2166 cm$^{-1}$). SRS-off images at 2000 cm$^{-1}$ showed negligible background with high vibrational specificity. Overlay images with fluorescent markers showed good co-localization. Protein CH$_3$ at 2940 cm$^{-1}$ was imaged to display cell morphology. **d** 6-color SRS imaging of live cells, including nucleus (EdU, 2116 cm$^{-1}$), plasma membrane (PM-switch, 2166 cm$^{-1}$), mitochondria (Mito-switch, 2205 cm$^{-1}$), lysosomes (Lyso-switch, 2232 cm$^{-1}$), protein CH$_3$ (2940 cm$^{-1}$), and lipid CH$_2$ (2845 cm$^{-1}$). Overlay of two images were shown for the same set of cells. Scale bar: 20 μm. All experiments were repeated three times independently with similar results.

## Reversible photoswitching and region-selective SRS imaging of live cells

Photoswitching imaging allows spatiotemporal selective molecular visualization with reversible light control. We performed photoswitchable SRS imaging in live cells using different organelle-targeted Carbow-switch (Fig. 5). Before 405 nm irradiation, SRS imaging of PM-switch showed a strong plasma membrane signal in the PM-open channel (2166 cm$^{-1}$), which outlined the cell morphology from the

protein methyl CH$_3$ channel (2940 cm$^{-1}$). Little signal was observed in the PM-closed channel (2148 cm$^{-1}$), demonstrating negligible frequency crosstalk and a clean background before photoswitching (Fig. 5a, b). After 405 nm illumination in the whole field of view, a bright plasma membrane signal appeared at 2148 cm$^{-1}$ by SRS imaging, which was consistent to the distribution in the PM-open channel and demonstrated efficient photoswitching in live cells. The signal can be completely switched off upon 640 nm irradiation, and this process of

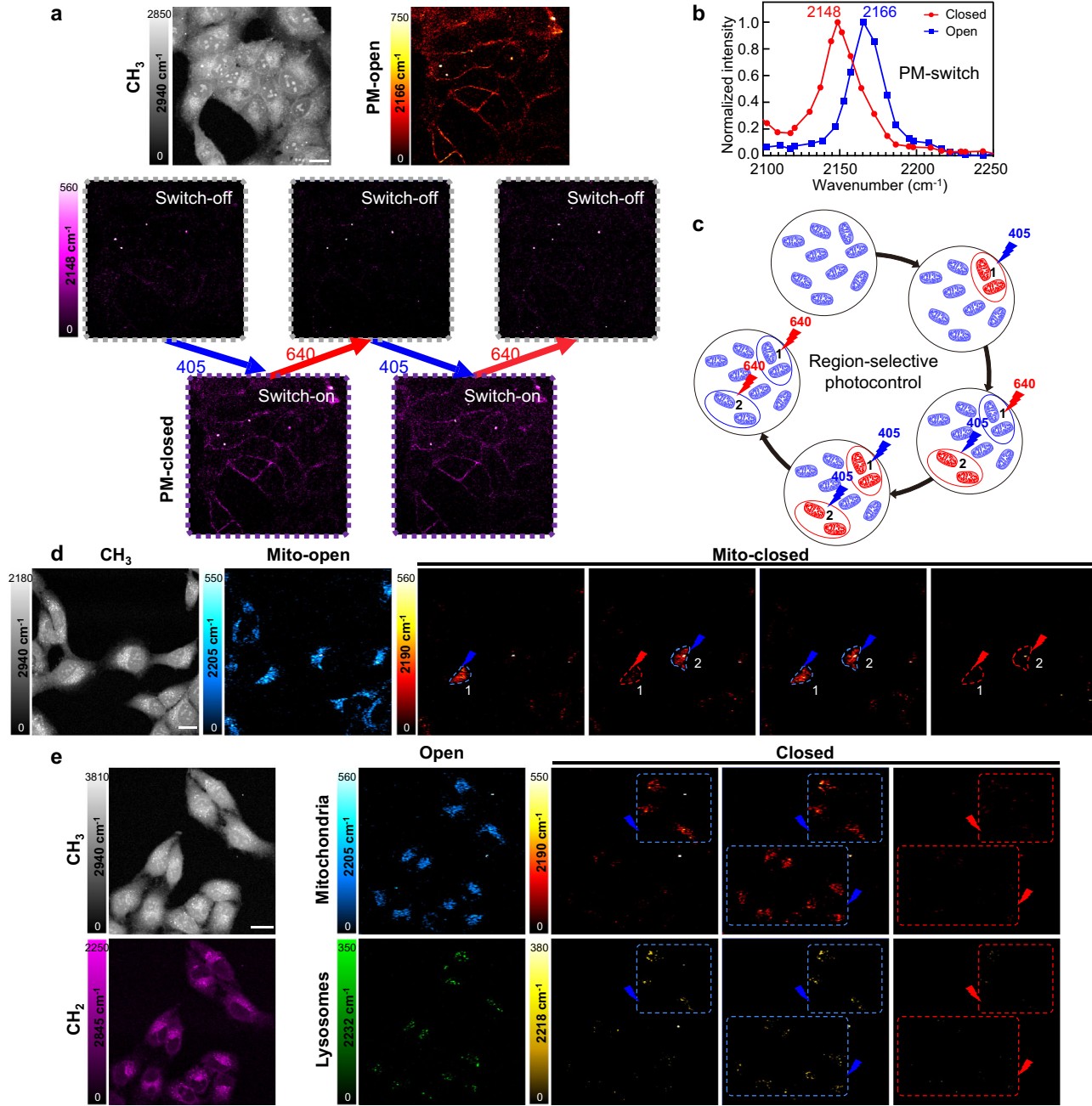

**Fig. 5 | Reversible photoswitching and region-selective SRS imaging of live cells with Carbow-switch. a** Reversible SRS imaging of PM-switch in live U2OS cells. Protein CH$_3$ (2940 cm$^{-1}$) and PM-open (2166 cm$^{-1}$) channels were imaged before photoswitching to show the cell morphology and plasma membrane staining. Using alternate 405 nm and 640 nm irradiations, SRS signal of plasma membrane at the PM-closed channel of 2148 cm$^{-1}$ can be reversible switched on and off for two cycles. **b** Normalized SRS spectra of PM-switch in the open and closed forms. **c** Scheme for region-selective photocontrol in live cells. Blue and red arrows indicated 405 nm and 640 nm irradiations in specified regions, respectively. **d** Region-selective photoswitchable SRS imaging of Mito-switch in live U2OS cells. Mitochondria signal in

region 1 and 2 (circled by blue or red dashed lines) were selectively modulated and imaged at the Mito-closed channels of 2190 cm$^{-1}$ using 405 nm and 640 nm lights. **e** Multi-region photoswitching and multiplexed SRS imaging of mitochondria and lysosomes in live U2OS cells. Mito-closed (2190 cm$^{-1}$) and Lyso-closed (2218 cm$^{-1}$) channels were imaged in two selected regions by photoswitchable SRS microscopy under 405 nm and 640 nm irradiations. Mito-open channel at 2205 cm$^{-1}$ and Lyso-open channel at 2232 cm$^{-1}$ were imaged to show the total distribution of mitochondria and lysosomes. Protein CH$_3$ (2940 cm$^{-1}$) and lipid CH$_2$ (2845 cm$^{-1}$) were imaged to show the cell morphology. Scale bar: 20 μm. All experiments were repeated three times independently with similar results.

on- and off-switching can be repeated for multiple times using two colors of visible light (Fig. 5a).

We also demonstrated reversible photoswitching and SRS imaging of Mito-switch and Lyso-switch inside living cells with low background and high contrast (Supplementary Fig. 13). Moreover, by using both Mito-switch and Lyso-switch, we have achieved multi-color reversible SRS imaging of live cells (Supplementary Fig. 14). In the 6-channel SRS images including protein $CH_3$, lipid $CH_2$, Mito-open, Mito-closed, Lyso-open and Lyso-closed, the signal of mitochondria and lysosomes can be reversibly switched on and off for two cycles and visualized at the corresponding channels of 2190 $cm^{-1}$ and 2218 $cm^{-1}$, respectively, which showcased both the well-resolved photoswitchable signal and the multiplexing capability of Carbow-switch.

Furthermore, we applied Carbow-switch for region-selective photocontrol and SRS imaging inside living cells. By irradiating 405 nm or 640 nm lights at specified areas, we can modulate the signal of Carbow-switch reversibly at the single cell level. We have demonstrated photoswitchable subcellular imaging of mitochondria using Mito-switch (Fig. 5c, d). Through selective 405 nm irradiation, we first switched on mitochondria in region 1 for SRS imaging inside a single cell, while the rest of cells didn't show any signal. Then, the signal of region 1 was switched off with 640 nm irradiation and mitochondria in region 2 was selectively switched on by 405 nm light. This allowed SRS visualization of specific organelles at different time. Both regions of mitochondria signal can be switched on again and imaged by SRS simultaneously. Afterwards, all mitochondria signal in the field of view was switched off by 640 nm irradiation for next round of region-selective photoswitchable imaging.

To show the versatility of photocontrol, we also demonstrated different modes of photoswitching in Mito-switch and Lyso-switch by changing the illumination area of visible lights, which included rectangular, circular, single-cell or other irregular shapes (Supplementary Figs. 15 and 16). Further, Multi-region photoswitching and multiplexed SRS imaging of mitochondria and lysosomes has been achieved in live cells. With both Mito-switch and Lyso-switch, different areas of mitochondria and lysosomes can be reversibly highlighted and selectively visualized within the same set of cells (Fig. 5e). Therefore, Carbow-switch has achieved multiplexed photoswitchable SRS imaging of organelles with reversible control and high specificity, which allowed spatiotemporal selective study of subcellular interactions and dynamics.

### Photo-controllable time-lapse imaging of organelle dynamics in live cells

Organelle organization and dynamics are key indicators of cellular states. Photo-selective time-lapse imaging of Carbow-switch was carried out to visualize specific organelle dynamics and interactions in live cells. Mitochondria is a central intracellular energy source through redox metabolism in the tricarboxylic acid cycle, and dysfunction of mitochondria is associated with many human diseases including diabetes and neurological disorders[43]. With Mito-switch, we tracked the mitochondria dynamics inside single living cell under both normal and oxidative conditions. U2OS cells, the human osteosarcoma epithelial cell line, were labeled with Mito-switch and selected mitochondria was switched on by 405 nm and imaged by time-lapse SRS microscopy (Fig. 6a). After incubation in phosphate buffered saline (PBS) for 10 min, the mitochondrial intensity in the same area was decreased to 84%, indicating active movement of mitochondria. The merged SRS images at two time points showed dispersed pattern with partial intensity overlap, suggesting dynamic mitochondrial network in live cells.

Hydrogen peroxide ($H_2O_2$) is a key signaling molecule in cells and high concentration of $H_2O_2$ induces cellular oxidative stress[44]. In live U2OS cells treated with 2 mM $H_2O_2$, we tracked the organization of selective mitochondria by time-lapse SRS imaging (Fig. 6b), which differed from the normal PBS condition. Compared to the dispersed distribution at 0 min, mitochondria were found to concentrate near the nucleus after 30 minutes of $H_2O_2$ treatment (Fig. 6b arrowhead), as shown by the strong signal and compact pattern in the merged image. Oxidative stress can lead to apoptosis through mitochondria-dependent pathway. The mitochondrial aggregation we observed was likely an early stage in the process of cytochrome c release during apoptosis[45].

Moreover, to demonstrate the photoswitching capability, we performed photoswitchable SRS imaging and tracking of subcellular dynamics and intercellular organelle trafficking in living cells. In U2OS cells, subpopulations of lysosomes and mitochondria were selectively highlighted and tracked inside single cells with high specificity, which clearly showed fast dynamics within live cells in 10 minutes (Supplementary Fig. 17). In mouse embryonic fibroblast (MEF) cells cultured on fibronectin-coated dishes[46], we applied photoswitchable imaging to visualize intercellular organelle trafficking, which showed large movements of lysosomes from one cell to the neighboring cells (Supplementary Fig. 18), suggesting active cell-cell communication[47].

Furthermore, we performed time-lapse imaging of Lyso-switch to study selective lysosome dynamics during stress granule formation (Fig. 6c). Stress granules are membraneless organelles assembled through protein liquid–liquid phase separation in response to different stress stimuli, which are crucial in the regulation of mRNA translation, degradation, and cell signaling[48]. Ataxin-2 protein is shown to be an important component of stress granules in mammalian cells, the mutation of which is implicated in the development of several neurodegenerative diseases including spinocerebellar ataxia type 2 (SCA2) and amyotrophic lateral sclerosis (ALS)[49]. Using U2OS cells stably transfected with EGFP-ATXN2[50], where the N terminal of ataxin-2 was fused to EGFP, we were able to track the organization and interactions of stress granules and lysosomes using correlated fluorescence and SRS microscopy.

Sodium arsenite ($NaAsO_2$) was used to induce the formation of stress granules[51]. Before $NaAsO_2$ incubation, ataxin-2 protein was evenly distributed in the cytoplasm and lysosomes displayed clear puncta signal inside living cells from multiplexed imaging of EGFP and Lyso-switch. We then incubated cells with 2 mM $NaAsO_2$ and highlighted lysosomes in selected cells using 405 nm light. After 30 min incubation, we found that most ataxin-2 condensed strongly into bright puncta inside the cytoplasm, indicative of stress granules (Fig. 6c thick arrow and Supplementary Fig. 19a). The distribution of lysosomes remained scattered and there is little overlap between lysosomes and stress granules, suggesting the absence of direct interaction. Interestingly, in cells with few stress granules, we observed that lysosomes formed large clusters in the perinuclear region, likely the microtubule-organizing center (Fig. 6c thin arrow). This may suggest an indirect regulation between stress granule formation and lysosomal clustering in cells under stress.

Although ataxin-2 is mainly expressed in the cytoplasm, we also found heterogeneous distribution of ataxin-2 inside the cell nucleus. In a few cells with nuclear localization of ataxin-2, there was little formation of stress granules with $NaAsO_2$ treatment (Supplementary Fig. 19b, c). At the same time, the lysosomes showed strong clustering with increased intensity. Together, these results suggested that lysosomes changed their intracellular positioning in cells without stress granules, which might regulate cellular homeostasis under stress by coordinating their interactions with autophagosomes via mTORC1 signaling[52,53]. Thus, through spatial-selective photocontrol and time-lapse multiplexed imaging of Carbow-switch, we have demonstrated its potentials for studying complex subcellular dynamics and interactions.

### Discussion

In this study, we have successfully developed a new palette of Carbow-switch through rational coupling of asymmetric DAE and polyynes,

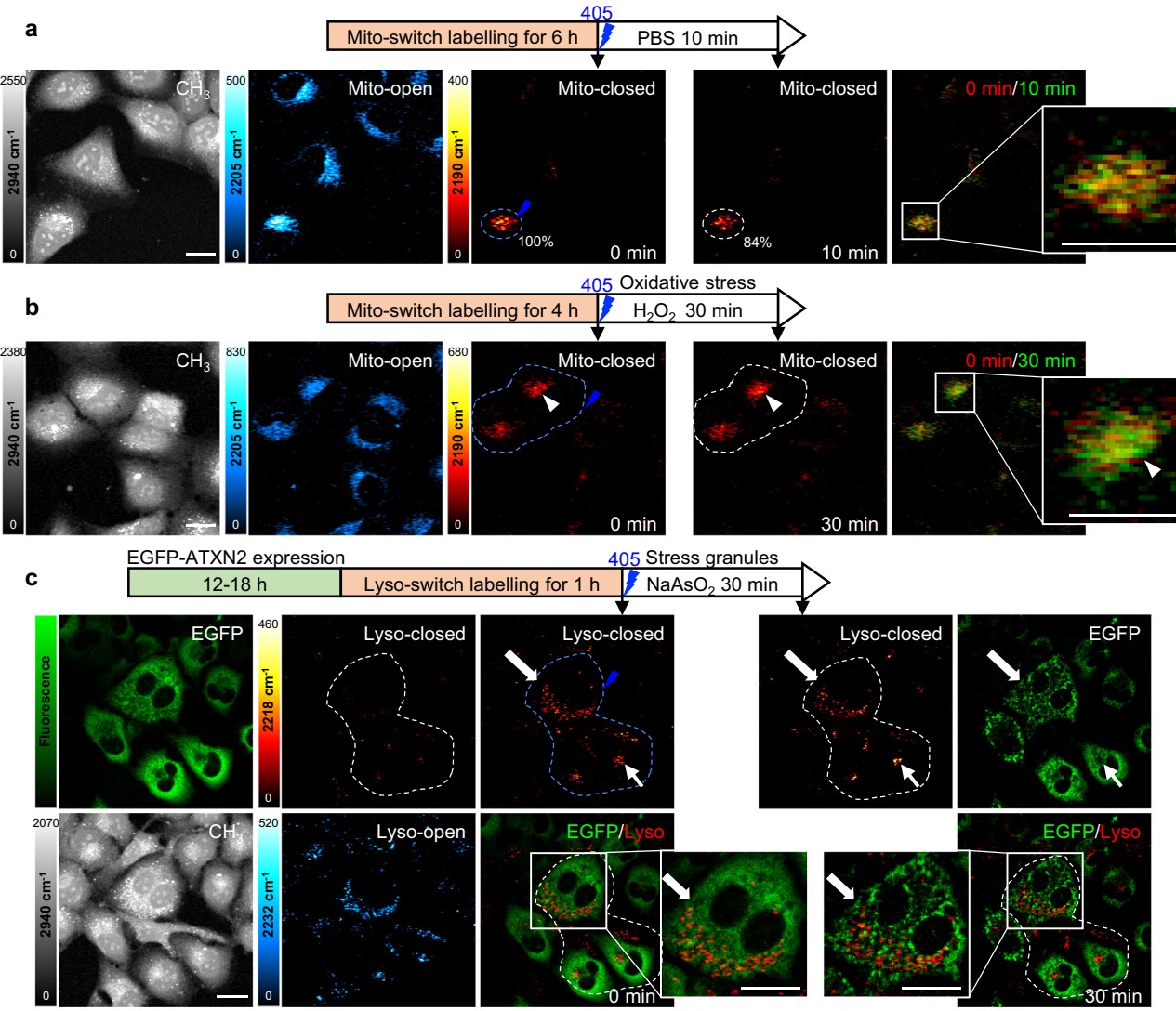

**Fig. 6 | Live-cell time-lapse imaging of selective organelle dynamics with Carbow-switch. a** Photo-selective mitochondrial imaging in normal condition. Live U2OS cells were labeled with Mito-switch and selectively switched on by 405 nm irradiation (dashed lines in Mito-closed channels). Time-lapse SRS imaging of selected mitochondria showed the decrease of intensity after 10 min in PBS. Overlay image of mitochondria at 0 min (red) and 10 min (green) showed the dispersed pattern with partial overlap at two time points. **b** Photo-selective mitochondrial imaging under oxidative stress. Live U2OS cells labeled with Mito-switch were selectively switched on by 405 nm irradiation (dashed lines in Mito-closed channels). Time-lapse SRS imaging of mitochondria in selected cells showed compact distribution with strong signal after 30 min treatment in $H_2O_2$. Overlay image at 0 min (red) and 30 min (green) indicated mitochondrial aggregation with time (white arrowhead). **c** Photo-selective lysosomal imaging during stress granule

formation. Live U2OS cells expressing EGFP-ATXN2 were labeled with Lyso-switch and selectively switched on by 405 nm irradiation (dashed lines in Lyso-closed channels). Time-lapse fluorescence imaging of EGFP showed bright puncta formation of stress granules in the cytoplasm during 30 min sodium arsenite treatment. Time-lapse SRS imaging of lysosomes showed heterogeneous distributions in selected cells. In cells with many stress granules, lysosomal distribution remained scattered (thick arrow). In cells with few stress granules, lysosomes formed clusters in the perinuclear region (thin arrow). Overlay image of selected lysosomes (red) and EGFP-ATXN2 (green) at both 0 min and 30 min showed little overlap between lysosomes and stress granules. All protein $CH_3$ were imaged by SRS to show the cell morphology. Scale bar: 20 μm. All experiments were repeated three times independently with similar results.

achieving 16 vibrational channels with excellent photoswitching properties for multiplexed photoswitchable SRS microscopy. First, the electronic spectra of asymmetric DAE polyynes were tuned precisely between 560 nm and 720 nm for near-IR epr-SRS excitation, through heterocycle and polyyne incorporation. Next, we greatly expanded the Raman frequencies of DAE polyynes by end-capping variation, $^{13}C$ isotope labeling, and polyyne elongation, generating 16 photoswitchable peaks in the cell Raman-silent window for multiplexed detection. Carbow-switch molecules displayed high photo-conversion yields (up to 93%), fast switching kinetics under low irradiation power (microwatt level), and excellent photo-fatigue resistance (95% after 100 cycles). More importantly, the SRS signal of all Carbow-switch can be reversibly

switched on and off using two visible lights, causing large frequency shift between open and closed forms (12–24 cm⁻¹) and strong intensity increase upon on-switching (up to 90-fold enhancements), which allowed photoswitchable SRS detection with high sensitivity, specificity, and multiplexing capability.

Moreover, we functionalized Carbow-switch for organelle-specific imaging, including mitochondria, lysosomes and plasma membrane, which was demonstrated in 6 and 9-channel photoswitchable SRS imaging of live cells. We further achieved reversible photoswitching and region-selective SRS imaging in living cells with high spatial photocontrol using different organelle-targeted Carbow-switch. Lastly, we selectively tracked subcellular dynamics of mitochondria and

lysosomes during oxidative stress, cell-cell communication, and protein phase separation using time-lapse multiplexed photoswitchable imaging of Carbow-switch, which revealed complex organization of organelles under cellular stress and signaling.

Carbow-switch can be improved for photoswitchable vibrational detection. Epr-SRS excitation efficiency closely depends on the molecular absorption[37]. Currently, the absorption coefficients of Carbow-switch in the closed forms are in the range of $10^4 M^{-1} cm^{-1}$ (Supplementary Table 1), which is moderate compared to strongly-absorbing molecules (in $10^5 M^{-1} cm^{-1}$). Sulfur oxidation of thiophene and site-specific substitution in DAE were shown to significantly increase the extinction coefficient and fluorescence quantum yield[54], which could lead to stronger epr-SRS signal enhancement due to the multiphoton process and facilitate single-molecule technique[55]. Recently, super-resolution SRS imaging based on photoswitchable DAE was demonstrated to enhance the spatial resolution by 2–4 folds[56,57]. With larger signal and multiplexed detection of Carbow-switch in the cell Raman-silent region, super-resolution SRS microscopy with greater resolution enhancement and multiplexing channels will be achievable. In summary, we have developed multiplexed photoswitchable SRS microscopy with Carbow-switch, which provides a powerful tool with great potentials for interrogating selective interaction and dynamics in living systems with high spatiotemporal control.

## Methods

### Chemical synthesis
Methods for chemical synthesis and characterization of all compounds can be found in the Supplementary Notes.

### Stimulated Raman scattering microscopy
Synchronized pump and Stokes laser beams at 80 MHz repetition rate were produced from an integrated laser system (picoEMERALD, Applied Physics and Electronics, Inc.). The Stokes laser beam (1032.0 nm, 2 ps pulse width, spectral bandwidth ~10 cm$^{-1}$) was intensity modulated at 10 MHz by an electro-optic-modulator with >90% modulation depth, and the pump laser beam (tunable between 700–990 nm, 2 ps pulse width, spectral bandwidth ~10 cm$^{-1}$) was generated by a built-in optical parametric oscillator. The two laser beams were spatially and temporally overlapped by using two dichroic mirrors and an integrated decay stage, and both beams were coupled into an inverted laser scanning microscope (Olympus, FV3000). The lasers were focused onto the sample through a 25× water objective (XLPLN25XWMP, 1.05 N.A., Olympus). After the sample, the transmitted beams were effectively collected by a high-NA condenser lens (oil immersion, 1.4 N.A., Olympus) and the Stokes beam was removed by a bandpass filter (ET890/220 m, CHROMA). The remaining pump beam was detected by a large area (10×10 mm) silicon photodiode (FDS1010, THORLABS) reverse biased at 64 DC voltage to maximize saturation threshold and reduce response time. The output current was terminated by a 50-Ω terminator and filtered with a 9.5–11.5 MHz bandpass filter (BBP-10.7+, Mini-Circuits) to reduce laser noise. The stimulated Raman loss signal was extracted by demodulation at the Stokes modulation frequency of 10 MHz using a lock-in amplifier (SR844, Stanford Research Systems or HF2LI, Zurich instrument). The in-phase signal was sent to the analog interface box (FV30-ANALOG, Olympus) of the microscope to generate SRS images.

All SRS images were acquired with a 30 μs time constant set by the lock-in amplifier and 40 or 80 μs pixel dwell time for a 320×320 or 512×512-pixel field of view unless otherwise specified. 14-43 mW of pump power and 65-87 mW of Stokes power measured after the objective were used for cell imaging. For photoswitchable imaging, 14-29 mW of pump power and 22-43 mW of Stokes power were used. And 664-830 μW of 405 nm and 485 μW of 640 nm CW lasers (Olympus, FV3000) were used to photo-switch selective regions with 40 μs pixel dwell time. Photoswitching kinetics were measured using

2 μs pixel dwell time. For photostability imaging of Lyso-switch, 7 μs time constant and 10 μs pixel dwell time were used. SRS spectra were acquired by scanning the pump laser wavelength. SRS spectrum of 10 μM 2o-4-OH was measured under 60 mW of pump power and 170 mW of Stokes power with 1 ms time constant. 10 mM EdU (TCI, E1057) in PBS was used as a standard for RIE measurements.

All fluorescence images were collected by the same confocal microscope (Olympus, FV3000) with CW laser excitation (405, 488, 561 and 640 nm) and standard bandpass filter sets. All images were analyzed and assigned color by ImageJ. In all SRS color bars, every 100 intensity corresponds to 0.92 μV SRS signal.

### UV-Vis absorption spectroscopy
UV-Vis absorption spectra were measured on a Cary 60 UV-Vis Spectrophotometer (Agilent). The 365 nm UV light for photoswitching was from a UV handheld lamp (MINGREN, ZF-5) with power of 5 W. The 405 nm and 640 nm visible lights were from two LEDs each with power of 6 W.

### Cell culture
U2OS (HTB-96, ATCC) and MEF cells (SCRC-1008, ATCC) were cultured in the DMEM medium (Hyclone, SH30022.01), supplemented with 10% fetal bovine serum (Gemini, 900-108) and 1% penicillin-streptomycin (Hyclone, SV30010) at 37 °C and 5% CO$_2$. Cells were seeded into 24-well plates on 14 mm glass coverslips (NEST, 801010) with a density of 60,000 to 80,000 cells per well for imaging experiments. *ATXN2/2L* double knockout U2OS cell line stably transfected with pCW-EGFP-ATXN2 plasmid (Tet-On system) was a generous gift from Prof. Yi Lin, and cultured in the same condition. All cell samples were washed with PBS for three times, and assembled into a chamber made of imaging spacer (iSpacer, IS216) filled with PBS solution before imaging unless otherwise specified.

### Cell viability assay
Cell viability assay was performed using Calcein-AM/PI live/dead cell staining kit (Solarbio, CA1630). U2OS cells treated with Carbow-switch probes were incubated with 2 μM Calcein-AM and 3 μM PI for 20 min at 37 °C before fluorescence imaging.

### Live-cell SRS and fluorescence imaging

### Mitochondria imaging
U2OS cells were cultured on glass coverslips and incubated with 20 μM Mito-switch and 0.1% Pluronic F-127 (Beyotime, ST501) for 4 h at 37 °C. For colocalization experiments, 100 nM MitoTracker Green FM (Cell Signaling Technology, 9074 S) was added for 30 min at 37 °C. For photoswitching experiments, U2OS cells were incubated with 40 μM Mito-switch and 0.1% Pluronic F-127 for 4 h at 37 °C. For photostability experiments, U2OS cells were incubated with 40 μM Mito-switch and 0.1% Pluronic F-127 for 6 h at 37 °C.

### Lysosomes imaging
U2OS cells were cultured on glass coverslips and incubated with 5 μM Lyso-switch and 0.1% Pluronic F-127 for 1 h at 37 °C. For colocalization experiments, 200 nM LysoTracker Red DND 99 (Meilunstar, MB6041-1) was used to stain cells for 30 min at 37 °C. For photoswitching experiments, U2OS cells were incubated with 8 μM Lyso-switch and 0.1% Pluronic F-127 for 1 h at 37 °C. For photostability experiments, U2OS cells were incubated with 10 μM Lyso-switch and 0.1% Pluronic F-127 for 1 h at 37 °C.

### Plasma membrane imaging
U2OS cells were cultured on glass coverslips and incubated with 40 μM PM-switch and 0.1% Pluronic F127 for 45 min at 37 °C. For colocalization experiments, 0.5 μg/mL CellMask Plasma Membrane Deep Red

(Invitrogen, C10046) was used to stain cells for 8 min at 37 °C. For photoswitching experiments, U2OS cells were incubated with 60 μM PM-switch and 0.1% Pluronic F127 for 45 min at 37 °C.

### Multiplexed SRS imaging of mitochondria, lysosomes, plasma membrane, and EdU in live cells

For 6-color and 9-color live-cell photoswitchable SRS imaging, U2OS cells were cultured on glass coverslips and incubated with 300 μM EdU for 15 h at 37 °C. On the day of imaging, the cells were incubated with 30 μM Mito-switch and 0.1% Pluronic F-127 for 3 h, 4 μM Lyso-switch for 35 min and 40 μM PM-switch for 35 min at 37 °C. Cells were washed with PBS for three times before imaging.

### Multiplexed reversible SRS imaging of mitochondria and lysosomes in live cells

U2OS cells were cultured on glass coverslips and incubated with 40 μM Mito-switch and 0.1% Pluronic F-127 for 4 h and 7 μM Lyso-switch for 1 h at 37 °C. Cells were washed with PBS for three times before imaging.

### Time-lapse photoswitchable SRS imaging of mitochondria and lysosomes in PBS

For time-lapse photoswitchable imaging of mitochondria, U2OS cells were cultured on glass coverslips and incubated with 40 μM Mito-switch and 0.1% Pluronic F-127 for 4-6 h at 37 °C. For time-lapse photoswitchable imaging of lysosomes, U2OS cells were cultured on glass coverslips and incubated with 10 μM Lyso-switch and 0.1% Pluronic F-127 for 1 h at 37 °C. Cells were washed with PBS for three times before imaging, and incubated with PBS for 10 min at room temperature in an imaging chamber for time-lapse imaging.

### Time-lapse photoswitchable SRS imaging of mitochondria under $H_2O_2$ treatment

U2OS cells were cultured on glass coverslips and incubated with 40 μM Mito-switch and 0.1% Pluronic F-127 for 4 h at 37 °C. Cells were washed with PBS for three times before imaging, and incubated with 2 mM $H_2O_2$ (Sinopharm Chemical Reagent, 10011218) in PBS for 30 min at room temperature in an imaging chamber for time-lapse imaging.

### Time-lapse photoswitchable SRS imaging and tracking of lysosomes in MEF cells

14 mm glass coverslips were pre-coated with 10 or 20 μg/mL fibronectin (MedChemExpress, HY-P3160) at 37 °C for 4 h. MEF cells were cultured on fibronectin-coated coverslips for 12-16 h. Cells were incubated with 7 μM Lyso-switch and 0.1% Pluronic F-127 in Opti-MEM (no phenol red, Gibco, 11058-021) for 1 h at 37 °C. Cells were washed with Opti-MEM once before imaging, and incubated in Opti-MEM for 30 min at room temperature in an imaging chamber for time-lapse SRS imaging.

### Time-lapse SRS and fluorescence imaging of lysosomes and stress granules under arsenite treatment

U2OS cells stably transfected with EGFP-ATXN2 were cultured on glass coverslips and incubated with 5 μM doxycycline (Bide Pharmatech, BD30058) to induce the expression of EGFP-ATXN2 for 12-18 h at 37 °C. On the day of imaging, cells were incubated with 10 μM Lyso-switch and 0.1% Pluronic F-127 for 1 h at 37 °C. Cells were washed with PBS for three times before imaging, and incubated with 2 mM NaAsO2 (Beijing Weiye Research Institution, BWR3038-2016) in HBSS for 30 min at 37 °C in an imaging chamber for time-lapse fluorescence and SRS imaging.

### Reporting summary

Further information on research design is available in the Nature Portfolio Reporting Summary linked to this article.

## Data availability

The data supporting the findings of this study are available within the paper and in Supplementary Figures, Supplementary Tables, and Supplementary Notes. All other data are available from the corresponding author upon request.

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

## Acknowledgements

This work was supported by the National Natural Science Foundation of China (22174085 to F.H.) and Tsinghua University Initiative Scientific Research Program. We thank Prof. Yi Lin for providing the cell strain.

## Author contributions

F.H. conceived the research. Y.Y. and F.H. designed the experiments. Y.Y. and X.B. completed the chemical synthesis and spectroscopic studies. Y.Y. and X.B. carried out live cell imaging studies and analyzed the data. F.H. supervised the project. Y.Y., X.B. and F.H. wrote the manuscript.

## Competing interests

The authors declare no competing interests.
