## [Peer Review File · Nature Communications]

Reviewers' Comments:

Reviewer #1:

Remarks to the Author:

Vibrational microscopy has demonstrated superior chemical specificity and multiplexed imaging capability to fluorescence-based microscopy with much narrower bandwidth. However, most existing Raman probes are inert, difficult to be controlled with external stimuli, hindering some fascinating functions such as sensing and manipulation. In 2021, photoswitchable SRS imaging has been demonstrated by multiple groups, including the use of diarylethene-alkyne compounds, and this work is a nice continuation by providing a palette of photoswitchable polynes based on diarylethene derivatives to exploit the multiplexing ability. Overall, the work is interesting, and the manuscript is well-written with adequate data support. Therefore, I would recommend its publication after appropriate revisions. Please find the comments listed below.

General comments:

1. Why do the authors choose the strategy of asymmetric design? Please clarify the reasons and advantages.
2. Fig. 6 and Fig. S13 present the organelle dynamics by time-lapse imaging, which could also be handled by "always on" Raman probes. Can the authors clarify the necessity of photoswitchable capability in this demonstration.
3. The multi-color imaging is demonstrated mainly by separately labeling different samples, which highly restricts the significance. Even if over 10-color probes are synthesized, they cannot be applied simultaneously (on the same sample), and thus the true multiplexing is severely weakened. Moreover, I'm wondering if multi-color labelling would harm the living cells? Maybe it's more suitable for tissues. Please provide more discussions on the multiplexing imaging applications.
4. Table S2 only provide 10 probes and the frequency of 2-1 and 2-2-OH is difficult to be distinguished. I suggest the authors give a table in details to summarize all the 16 color photoswitchable Raman frequencies to avoid overclaims.

Specific comments:

1. Line 50-51, I'm a little confuse with the interpretation of previous work of ref 34 and 35. Ref 34 shows more than single-color imaging in their work already, and the molecule used in ref 35 does not seem to be dithienylethene. Hope the authors could verify these to give correct statements.
2. What's the spectral resolution of the SRS setup. The FWHM in Fig. 2 and 5 are broad, which might reduce the on-off-ratio.
3. In the experiment of photo-fatigue resistance (Fig. S2), it would be better to use Raman as the readout signal rather than absorption.
4. Why is fluctuation in Fig. S4 so high? How's the stability of the system?
5. Please give the calibration bar of LUT in the figures.
6. Please give calculation details for the 300 nM detection limit.
7. The authors might want to cite a new work on super-resolution SRS microscopy based on similar photoswitchable probes [Adv. Photon. 5(6), 066001 (2023), doi: 10.1117/1.AP.5.6.066001.]

Reviewer #2:

Remarks to the Author:

The manuscript reports on the development of photo-switchable Carbow SRS probes dubbed "switch-carbow". The results are convincing and are a significant step forward in the field. The data shown are supporting the conclusions made, and I believe the results are significant enough to be suited for publication in Nature Communications.

I however have some comment which the authors should address before possible publication:

- 1) The use of "for the first time" claims should be avoided – it is understood that the results are new, and it is difficult to verify this claim.
- 2) Small language issues in the text "... increasing attentions in biological..." -> attention; "...signal, and has achieved 3-color." "have been used for..."; "To achieve multiplexed SRS imaging with selective photocontrol will enable" – remove "to achieve"; ...
- 3) "we measured the absorption intensity" – what is that ? absorption is reported as optical density

over a given pathlength, or otherwise – please define what is shown.

4) Use of 2-1 and 3-1 and 3c-1 and 3o-1 etc in the text is unclear – please define your nomenclature explicitly.

5) I believe an important information missing from the work is the characterisation of photostability of the probes under SRS imaging and switching by the SRS excitation itself. This should be discussed clearly with experimental evidence, for example showing long term time-course imaging of the cells labelled with the different markers, and importantly in both switch states.

6) The use of the red and blue arrows in the images is confusing, as they cover part of the image data, and also look themselves similar to the imaged structures. Maybe better just to color the dashed lines encircling the exposed regions?

7) Fig.4: clarify explicitly in which switch states the data is taken.

8) Generally, in SRS data shown (Fig. 3,4,5,6): give quantitative SRS signal scales, best in relative modulation units. Clarify if any offset was subtracted.

Reviewer #3:

Remarks to the Author:

The manuscript by Fanghao Hu developed a palette of reversible photoswitchable SRS probes for multiplex detection. Spatial-selective multiplexed SRS imaging of different organelles was demonstrated in living cells. This method has great potential for multiplex living cell imaging. However, the authors do not fully demonstrate the potential of this great tool. Region-selective SRS imaging is not a killing application for this method. Many other easier methods can achieve region-selective imaging. Therefore, I will recommend publishing this paper after addressing the following points:

1. Page 5 Line 151. The authors claim that the detection limit is 300 nM when signal noise ratio =1. For reliable measurements, the minimum S/N should be more than 3. In addition, authors should provide the SRS spectrum at the detection limit concentration.
2. The authors design the Carbow-switch probes with low spectral crosstalk between its open-closed forms. While, for multiplex detection, the spectral crosstalk between open-closed forms of different probes should be considered as well. For example, the closed form of 2-1 (2194 cm⁻¹) has the same frequency as the open form of 2-2NH₂ (2194 cm⁻¹) and the peak of the closed form of 2-2NH₂ at 2170 cm⁻¹ is too close to the peak of the open form of 2-3-OH (2166 cm⁻¹). This crosstalk issue affects the multiplex ability of this method.
3. In Figure 4d, the authors only present the switch-on SRS imaging of live cells. While authors did not demonstrate the switch-off SRS imaging of live cells in every frequency channel.
4. The region-selective SRS based on Carbow-switch probes is interesting, but I do not see its killing bioimaging application. If researchers do not want to monitor specific regions, the easiest way is no laser irradiation at selected regions. If researchers want to observe specific organelles, the original multiplex epr-SRS probes have already been achieved.
5. All figures do not have color bars.

Photoswitchable polyynes for multiplexed stimulated Raman scattering microscopy with reversible light control

Reviewer #1:

Vibrational microscopy has demonstrated superior chemical specificity and multiplexed imaging capability to fluorescence-based microscopy with much narrower bandwidth. However, most existing Raman probes are inert, difficult to be controlled with external stimuli, hindering some fascinating functions such as sensing and manipulation. In 2021, photoswitchable SRS imaging has been demonstrated by multiple groups, including the use of diarylethene-alkyne compounds, and this work is a nice continuation by providing a palette of photoswitchable polyynes based on diarylethene derivatives to exploit the multiplexing ability. Overall, the work is interesting, and the manuscript is well-written with adequate data support. Therefore, I would recommend its publication after appropriate revisions. Please find the comments listed below.

We thank the reviewer for the positive evaluation and for recognizing the key advances in our work. Indeed, our study presents a new development of photoswitchable multiplexing palette for live-cell SRS imaging. We have addressed all the comments with additional data and discussion as below.

General comments:

1. Why do the authors choose the strategy of asymmetric design? Please clarify the reasons and advantages.

We thank the reviewer for bringing up this point. The asymmetric design of photoswitchable polyyne is crucial for its excellent optical properties. With the asymmetric structure of diarylethene, we can separately optimize the electronic and vibrational spectroscopy of photoswitchable polyynes.

First, asymmetric diarylethene allowed the incorporation of different heterocycles at one side to tune the absorption spectra further into near-infrared region as shown in Figure 1b. Compared to the commonly used thiophene, electron-donating methylindole ring can strongly redshift the absorption maxima of both open and closed forms through the push-pull effect. As a result, photoswitchable polyynes can be switched on with 405 nm visible light instead of UV

light. In addition, at the closed form, electron pre-resonance effect with 600-700 nm near-infrared absorption can enhance the SRS signal upon photoswitching.

Second, in asymmetric design, polyynes can be coupled into diarylethene independently, while maintaining the optimized absorption spectra. At the same time, the vibrational spectra can be tuned by modulating polyynes with different lengths, substitutions, and isotope labeling, which achieves vibrational multiplexing. If symmetric diarylethene is used, the tuning of electronic and vibrational spectra will be convoluted, which is highly limited. Moreover, the incorporation of polyynes at both sides could complicate the vibrational spectra further. Therefore, the design and development of asymmetric diarylethene polyynes allowed us to achieve both optimal electronic and vibrational spectroscopy for excellent photoswitching and multiplexing properties. We have now clarified the strategy of asymmetric design more in the main text (highlighted in red) to emphasize its key advantages.

2. Fig. 6 and Fig. S13 present the organelle dynamics by time-lapse imaging, which could also be handled by “always on” Raman probes. Can the authors clarify the necessity of photoswitchable capability in this demonstration.

With time-lapse photoswitchable SRS imaging in Fig. 6 and Fig. S13 (currently Fig. S19), we can track selected organelle dynamics in cellular regions or specific cells without signal interference from other areas of the cell or neighboring cells. If using “always on” probes for dynamic imaging, it will be challenging to distinguish the signal of interest from the non-selected background signal after certain time, especially when the organelles are widespread throughout the cells. Thus, photoswitchable imaging allowed us to encode spatiotemporal information into different frequencies and highlight organelles in selective regions or cells for dynamic tracking with high specificity.

To further demonstrate the necessity of photoswitching capability, we performed photoswitchable SRS imaging and tracking of subcellular dynamics and intercellular organelle trafficking in living cells. In U2OS cells, subpopulations of lysosomes and mitochondria were selectively highlighted and tracked inside single cells with high specificity, which clearly showed fast dynamics within live cells in 10 minutes and was difficult to identify with “always on” Raman probes. In MEF cells cultured on fibronectin-coated dishes, we applied photoswitchable imaging to visualize intercellular organelle trafficking, which showed the

movements of lysosomes from one cell to the neighboring cells, suggesting active cell-cell communication. Therefore, both experiments showcased the necessity and advantages of photoswitchable SRS imaging compared to the “always on” probes. We have now updated the main text and added the new data in the SI as Fig. S17 and Fig. S18.

Subcellular photoswitchable imaging and time-lapse tracking of lysosomes and mitochondria inside living cells. Scale bar: 20 μm .

Photoswitchable SRS imaging and tracking of lysosomes among MEF cells. Scale bar: 20 μm .

3. The multi-color imaging is demonstrated mainly by separately labeling different samples, which highly restricts the significance. Even if over 10-color probes are synthesized, they cannot be applied simultaneously (on the same sample), and thus the true multiplexing is severely weakened. Moreover, I'm wondering if multi-color labelling would harm the living cells?

In Figures 4d and 5e, we have labelled the same cell samples with different organelle-targeted Carbow-switch probes simultaneously, including mitochondria, lysosome, and plasma membrane as well as EdU, which demonstrated the feasibility of multi-color labeling and imaging in the same sample.

To further demonstrate the multiplexing capability with Carbow-switch, we performed 9-color SRS imaging of the same living cells in both open and closed photoswitching forms, including lysosomes Lyso-switch (open and closed frequencies, 2232 and 2218 cm^{-1}), mitochondria Mito-switch (open and closed frequencies, 2205 and 2190 cm^{-1}), plasma membrane PM-switch (open and closed frequencies, 2166 and 2148 cm^{-1}), nucleus (EdU, 2116 cm^{-1}), total protein CH_3 (2940 cm^{-1}), and total lipids CH_2 (2845 cm^{-1}). We have updated the main text and added this new data in SI as Fig. S12.

Live-cell multiplexed SRS imaging of organelles with Carbow-switch at both open and closed photoswitching forms. Overlay of two images were shown for the same set of cells. Scale bar: 20 μm .

Furthermore, with optimized incubation concentrations and time, multi-color labeling with Carbow-switch is fully compatible with living cells. We have performed cell viability assay under multiplexed Carbow-switch labeling condition, which showed little toxicity in live cells. We have now added this data to Fig. S10 in the SI.

Viability assay of organelle-targeted Carbow-switch in live U2OS cells. The same cells were labelled with EdU, Mito-switch, Lyso-switch and PM-switch probes simultaneously.

Maybe it's more suitable for tissues. Please provide more discussions on the multiplexing imaging applications.

We thank the review for this suggestion. We also agreed that tissue imaging with Carbow-switch is another promising direction for multiplexing application, which can be further combined with bead encapsulation. We are pursuing this direction in future work.

4. Table S2 only provide 10 probes and the frequency of 2-1 and 2-2-OH is difficult to be distinguished. I suggest the authors give a table in details to summarize all the 16 color photoswitchable Raman frequencies to avoid overclaims.

We thank the reviewer for this suggestion. In table S2, we summarized all 10 photoswitchable polyene compounds with detailed SRS characterizations, because they are newly reported molecules. Yes, the frequency of 2-1 and 2-2-OH are indeed similar, so we only selected 2-1 for the multiplexed Carbow-switch shown in Figure 3. We have now updated Table S2 to separately display the open and closed frequencies for each molecule and highlighted the selected Carbow-switch molecules of Figure 3 in bold to be more clear.

Specific comments:

1. Line 50-51, I'm a little confuse with the interpretation of previous work of ref 34 and 35. Ref 34 shows more than single-color imaging in their work already, and the molecule used in ref 35 does not seem to be dithienylethene. Hope the authors could verify these to give correct statements.

We thank the reviewer for the careful reading and have now modified the description of ref 34 to dual-color imaging in the main text.

In ref 35, the molecule used (shown below) is named as cis-1,2-dicyano-1,2-bis(2,4,5-trimethyl-3-thienyl)ethene, showing its structure of an ethene substituted by two thienyl groups. So it should belong to symmetric dithienylethene.

Structure of cis-1,2-dicyano-1,2-bis(2,4,5-trimethyl-3-thienyl)ethene in ref 35.

2. What's the spectral resolution of the SRS setup. The FWHM in Fig. 2 and 5 are broad, which might reduce the on-off-ratio.

The spectral resolution of our SRS setup is 14 cm^{-1} , which is determined by the 2-ps laser pulses. The spectra in Fig. 2 and 5 have a large frequency shift of 19 cm^{-1} between open and closed forms, which is significant compared to the FWHM of 21 cm^{-1} . The on-off ratio is measured to be as large as 31.8, which can be further enhanced with higher spectral resolution.

3. In the experiment of photo-fatigue resistance (Fig. S2), it would be better to use Raman as the readout signal rather than absorption.

We thank the reviewer for this advice. We have now performed photo-fatigue resistance experiment using SRS signal as the readout. With alternating 405 nm and 640 nm irradiations, over 90% SRS signal remained after 50 times of photoswitching. The result is consistent with that of Fig. S2, demonstrating high photo-fatigue resistance. We have now added this data in SI as Fig. S3.

Reversible photoswitching and photo-fatigue resistance of 2-3-OH over 50 times of alternating 405 nm and 640 nm illuminations. SRS intensity was measured at 2147 cm^{-1} .

4. Why is fluctuation in Fig. S4 so high? How's the stability of the system?

We have now repeated this experiment under the same condition and averaged over 10 independent measurements. The result is consistent with the previous one with much lower fluctuation, showing fast photoswitching under microwatt-level power. This also demonstrated the high stability of our system. We have updated this data in Fig. S6 of SI.

Photoswitching kinetics of 2-3-OH under 405 nm irradiations of different power. Each curve was averaged over 10 independent measurements.

5. Please give the calibration bar of LUT in the figures.

We have now added the calibration bars of LUT in the figures for all SRS images.

6. Please give calculation details for the 300 nM detection limit.

We calculated the detection limit as follows. The SRS calibration curve of 2c-4-OH (Fig. S4) was measured with 2.6 mW pump and 15 mW Stokes power on sample. The slope of the linear curve corresponded to a signal of 0.36 μV for 100 μM 2c-4-OH. SRS signal is proportional to the product of pump and Stokes power. Scaling to the higher 60 mW pump and 170 mW Stokes power allowed in our system, the signal was calculated to be 94 μV for 100 μM solution. At 60 mW pump power, the noise was measured to be 0.32 μV at 1 ms time-constant. So the S/N=1 detection limit = $0.32 \mu\text{V} / 94 \mu\text{V} * 100 \mu\text{M} \sim 340 \text{ nM}$.

We have also experimentally measured the peak intensity of 5 μM 2c-4-OH to be 2.3 μV under the same power condition. With the measured noise of 0.32 μV , the experimentally determined S/N=1 detection limit = $0.32 \mu\text{V} / 2.3 \mu\text{V} * 5 \mu\text{M} \sim 700 \text{ nM}$, which is close to the calculated detection limit. We have now modified the statement to emphasize more the relative intensity versus EdU (RIE) in the main text.

7. The authors might want to cite a new work on super-resolution SRS microscopy based on similar photoswitchable probes [Adv. Photon. 5(6), 066001 (2023), doi: 10.1117/1.AP.5.6.066001.]

We thank the reviewer for this suggestion and have now cited this new work as ref. 57 in the main text.

Reviewer #2:

The manuscript reports on the development of photo-switchable Carbow SRS probes dubbed “switch-carbow”. The results are convincing and are a significant step forward in the field. The data shown are supporting the conclusions made, and I believe the results are significant enough to be suited for publication in Nature Communications. I however have some comment which the authors should address before possible publication:

We thank the reviewer for the highly positive evaluation and pointing out that our development of multiplexed photoswitchable SRS imaging probes is a significant step forward in this field. We have addressed all the comments as below.

1. The use of “for the first time” claims should be avoided – it is understood that the results are new, and it is difficult to verify this claim.

We have now removed “for the first time” in the main text.

2. Small language issues in the text “... increasing attentions in biological...” -> attention; “..signal, and has achieved 3-color..” “have been used for..”; “To achieve multiplexed SRS imaging with selective photocontrol will enable” – remove “to achieve”; ...

We thank the review for the careful reading. We have now addressed all these issues in the main text.

3) “we measured the absorption intensity” – what is that ? absorption is reported as optical density over a given pathlength, or otherwise – please define what is shown.

We thank the reviewer for this correction. We have now changed “absorption intensity” to “absorption” in the text. In Figure 2d and Fig. S2, the data shown were normalized absorbance. We have also modified the “absorption intensity” in y axes of Figure 2b and Fig. S8 to “absorbance”.

4) Use of 2-1 and 3-1 and 3c-1 and 3o-1 etc in the text is unclear – please define your nomenclature explicitly.

We have now defined the nomenclature of photoswitchable polyynes more explicitly in the main text. The first number represents the heterocycles of asymmetric diarylethene, where 1, 2, 3, and 4 each represents benzothiophene, 1-methylindole, 5-methoxy-1-methylindole, and 5-dimethylamino-1-methylindole, respectively. The second number represents the number of triple bonds in the polyyne conjugated to the thiophene of asymmetric diarylethene, ranging

from 1 to 4. The letter in between represents the open (o) or closed (c) photoswitching form. Take 2c-2 as an example, it represents a structure of asymmetric diarylethene polyynes at the closed form with 1-methylindole heterocycle and two triple bonds in the polyyne.

5) I believe an important information missing from the work is the characterisation of photostability of the probes under SRS imaging and switching by the SRS excitation itself. This should be discussed clearly with experimental evidence, for example showing long term time-course imaging of the cells labelled with the different markers, and importantly in both switch states.

We thank the reviewer for this suggestion. We have now performed the photostability experiments of organelle-targeted Carbow-switch in live cells by continuous SRS imaging. SRS intensity of both Mito-switch and Lyso-switch kept steady over 95% after continuous 50-frame imaging in the open form, and in the closed form, over 80% signal of Lyso-switch remained after continuous time-course imaging of 20 frames, demonstrating good photostability of Carbow-switch probes in living cells. We have now added this data in the SI as Fig. S11.

Photostability characterization of Mito-switch and Lyso-switch by live-cell SRS imaging. SRS intensity of both Mito-switch and Lyso-switch in the open form stayed steady over 95% after continuous 50-frame imaging and in the closed form over 80% signal of Lyso-switch remained after continuous 20-frame imaging. Scale bar: 20 μm .

6) The use of the red and blue arrows in the images is confusing, as they cover part of the image data, and also look themselves similar to the imaged structures. Maybe better just to color the dashed lines encircling the exposed regions?

We have now reduced the size of red and blue arrows in Figures 5 and 6 so that it doesn't cover the key image data. We have also colored the dashed lines accordingly.

7) Fig.4: clarify explicitly in which switch states the data is taken.

The data in Figure 4 was taken in the open form of Carbow-switch. We have now added the open state in Figure 4 and the figure caption. To further demonstrate the multiplexed photoswitching capability with Carbow-switch, we performed 9-color SRS imaging of living cells in both open and closed photoswitching states, including lysosomes Lyso-switch (open and closed frequencies, 2232 and 2218 cm^{-1}), mitochondria Mito-switch (open and closed frequencies, 2205 and 2190 cm^{-1}), plasma membrane PM-switch (open and closed frequencies, 2166 and 2148 cm^{-1}), nucleus (EdU, 2116 cm^{-1}), total protein CH_3 (2940 cm^{-1}), and total lipids CH_2 (2845 cm^{-1}). Both photoswitching states were labeled as below. We have updated the main text and added this new data in SI as Fig. S12.

Live-cell multiplexed SRS imaging of organelles with Carbow-switch at both open and closed photoswitching forms. Overlay of two images were shown for the same set of cells. Scale bar: 20 μm .

8) Generally, in SRS data shown (Fig. 3,4,5,6): give quantitative SRS signal scales, best in relative modulation units. Clarify if any offset was subtracted.

We thank the reviewer for this suggestion. We have now added quantitative SRS signal scales in all SRS images. In Figure 3, the relative intensity of each molecule was summarized as relative SRS intensity versus EdU (RIE) in Supplementary Table 2. For all the color bars, every 100 intensity corresponds to 0.92 μV SRS signal. And no off-resonance image was subtracted in all data.

Reviewer #3:

The manuscript by Fanghao Hu developed a palette of reversible photoswitchable SRS probes for multiplex detection. Spatial-selective multiplexed SRS imaging of different organelles was demonstrated in living cells. This method has great potential for multiplex living cell imaging. However, the authors do not fully demonstrate the potential of this great tool. Region-selective SRS imaging is not a killing application for this method. Many other easier methods can achieve region-selective imaging. Therefore, I will recommend publishing this paper after addressing the following points:

We thank the reviewer for the positive evaluation, which appraised our method with great potential in multiplexed live-cell imaging. We have now addressed all the following points with additional data and discussion.

1. Page 5 Line 151. The authors claim that the detection limit is 300 nM when signal noise ratio =1. For reliable measurements, the minimum S/N should be more than 3. In addition, authors should provide the SRS spectrum at the detection limit concentration.

We thank the review for this suggestion. We have now experimentally measured the SRS spectrum of 2o-4-OH at a low concentration of 10 μ M on our system, which showed a good signal to noise ratio of 8. This translated to a S/N=1 detection limit of 1.3 μ M at 1 ms time constant. The result is higher than the calculated 300 nM, which is partly due to the signal enhancement from open to closed form. We agreed that it is more suitable to provide the SRS spectrum with S/N>3, so we have modified the statement in the main text to be “SRS spectrum of 2o-4-OH can be measured at 10 μ M with a good signal to noise ratio of 8 under 1 ms time constant”, and added this new spectral data in SI as Fig. S5.

SRS spectrum of 10 μ M 2o-4-OH in DMSO solution was measured with a signal to noise ratio of 8 at 1 ms time constant. The curve was averaged over 3 measurements.

2. The authors design the Carbow-switch probes with low spectral crosstalk between its open-closed forms. While, for multiplex detection, the spectral crosstalk between open-closed forms of different probes should be considered as well. For example, the closed form of 2-1 (2194 cm^{-1}) has the same frequency as the open form of 2-2NH₂ (2194 cm^{-1}) and the peak of the closed form of 2-2NH₂ at 2170 cm^{-1} is too close to the peak of the open form of 2-3-OH (2166 cm^{-1}). This crosstalk issue affects the multiplex ability of this method.

Indeed, it is more challenging to develop multiplexing frequencies for photoswitchable probes each with both open and closed forms. The spectral crosstalk among open forms, closed forms, and open-closed forms all needs to be considered. To resolve the crosstalk of 2-2-NH₂, 2-2-NH₂ might be removed from Figure 3, so that all of the remaining 14 frequencies are well resolved. In this case, the frequency difference between open-closed forms of different probes will be larger than 12 cm^{-1} for all Carbow-switch molecules as shown below. Although there is certain crosstalk of 2-2-NH₂, we think it is better to keep this molecule with phenyl amine substitution in the Carbow-switch palette of Figure 3 and researchers can select desired frequencies based on their specific needs. Also, more Carbow-switch frequencies could be further generated by increasing the polyene length (such as from 4 to 6) and isotope labeling.

3. In Figure 4d, the authors only present the switch-on SRS imaging of live cells. While authors did not demonstrate the switch-off SRS imaging of live cells in every frequency channel.

We thank the reviewer for this suggestion. We have now demonstrated 9-channel photoswitchable SRS imaging of live cells in both open and closed forms, including lysosomes Lyso-switch (open and closed frequencies, 2232 and 2218 cm^{-1}), mitochondria Mito-switch (open and closed frequencies, 2205 and 2190 cm^{-1}), plasma membrane PM-switch (open and closed frequencies, 2166 and 2148 cm^{-1}), nucleus (EdU, 2116 cm^{-1}), total protein CH_3 (2940 cm^{-1}), and total lipids CH_2 (2845 cm^{-1}). The switch-off and switch-on images were indicated as below. We have updated the main text and added this new data in SI as Fig. S12. We have also demonstrated multiplexed reversible SRS imaging of organelles in live cells, which showed the signal was switched on and off for two cycles in Fig. S14 of SI.

Live-cell multiplexed SRS imaging of organelles with Carbow-switch at both open and closed photoswitching forms. Overlay of two images were shown for the same set of cells. Scale bar: 20 μm .

4. The region-selective SRS based on Carbow-switch probes is interesting, but I do not see its killing bioimaging application. If researchers do not want to monitor specific regions, the easiest way is no laser irradiation at selected regions. If researchers want to observe specific organelles, the original multiplex epr-SRS probes have already been achieved.

Region-selective photoswitchable SRS imaging is particularly important for time-lapse dynamic tracking. There are active movements and transport of organelles between subcellular regions and between cells. With time-lapse photoswitchable SRS imaging, only selected organelles at a given region and time point will be switched on with a different frequency color and tracked in the whole field of view across different regions or cells, without signal interference from other areas of the cell or neighboring cells. This cannot be achieved by laser irradiation and imaging at selected regions only.

And if using the “always on” organelle probe for dynamic imaging, it will be difficult to distinguish the signal of interest from the non-selected background signal after certain time, especially when the organelles are widespread throughout the cells. Thus, in addition to regional selection, photoswitchable imaging allowed us to encode spatiotemporal information into different frequencies and highlight selective organelles in subcellular regions or cells for time-lapse dynamic tracking with high specificity.

To further demonstrate the necessity of region-selective photoswitching capability, we have newly performed photoswitchable SRS imaging and tracking of subcellular organelle dynamics and intercellular organelle trafficking in living cells (shown below). In U2OS cells, subpopulations of lysosomes and mitochondria were selectively highlighted and tracked inside single cells with high specificity, which clearly showed fast dynamics within live cells in 10 minutes and was difficult to identify with “always on” organelle probes.

In MEF cells cultured on fibronectin-coated dishes, we applied photoswitchable imaging to visualize intercellular organelle trafficking, which showed the movements of lysosomes from one cell to the neighboring cells, suggesting active cell-cell communication. Therefore, both experiments showcased the necessity and advantages of region-selective photoswitchable SRS imaging. We have now updated the main text and added the following new data in SI as Fig. S17 and Fig. S18.

Subcellular photoswitchable imaging and time-lapse tracking of lysosomes and mitochondria inside living cells. Scale bar: 20 μm .

Photoswitchable SRS imaging and tracking of lysosomes among MEF cells. Scale bar: 20 μm .

5. All figures do not have color bars.

We thank the reviewer for this suggestion. We have now added color bars in all figures for SRS images.

Reviewers' Comments:

Reviewer #1:

Remarks to the Author:

The authors have nicely addressed my previous concerns in the revision with additional supporting data. Hence I'd like to recommend the acceptance of the manuscript.

Reviewer #2:

Remarks to the Author:

The The authors have answered to my comments constructively and made corresponding changes in the manuscript. They have also answered to the comments of the other two reviewers in a similar fashion. I believe the work is suited for publication in Nature Communications.

Reviewer #3:

Remarks to the Author:

This revision has significantly improved the manuscript. I would like to recommend the manuscript for publication.